# CRISPRoff enables spatio-temporal control of CRISPR editing

Jared Carlson-Stevermer [1✉], Reed Kelso [1,2], Anastasia Kadina[1], Sahil Joshi [1], Nicholas Rossi[1], John Walker[1], Rich Stoner [1] & Travis Maures[1]

Following introduction of CRISPR-Cas9 components into a cell, genome editing occurs unabated until degradation of its component nucleic acids and proteins by cellular processes. This uncontrolled reaction can lead to unintended consequences including off-target editing and chromosomal translocations. To address this, we develop a method for light-induced degradation of sgRNA termed CRISPRoff. Here we show that light-induced inactivation of ribonucleoprotein attenuates genome editing within cells and allows for titratable levels of editing efficiency and spatial patterning via selective illumination.

---

[1] Synthego Corporation, 3565 Haven Avenue, Menlo Park, CA 94025, USA. [2]Present address: Freenome, 259 East Grand Ave, South San Francisco, CA 94080, USA. ✉email: jared.carlson-stevermer@synthego.com

CRISPR-Cas9 technology has rapidly revolutionized the genome editing field[1]. However, once introduced into a cell the CRISPR ribonucleoproteins (RNPs) are uncontrollable, capable of forming double strand breaks (DSBs) at locations where the sgRNA binds within the genome[2]. Previous attempts to control CRISPR systems inside the cell have approached this problem on a variety of levels, from controlled expression of integrated Cas9[3,4] and sgRNA[5], sgRNAs that are modulated in response to ligand binding[6], split Cas9 that dimerizes upon illumination with blue light[7], to anti-CRISPR proteins[8,9], a viral defense against the CRISPR immune system. However, each of these strategies requires an additional physical component of the CRISPR system beyond a typical RNP to be introduced into the cell. These additional engineering steps can also produce challenges to the production of purification of protein *Streptococcus pyogenes* Cas9 (SpCas9).

To address these challenges, we develop the CRISPRoff system, a synthetic sgRNA that fragments in response to light, preventing formation of new DSBs (Fig. 1a). Here we show that CRISPRoff is effective across multiple genomic targets in multiple cell lines and demonstrate two key uses of this platform; the ability to maximize on:off targeting editing events and the ability to spatially pattern cells in vitro.

## Results

**CRISPRoff sgRNA synthesis and cleavage.** CRISPRoff sgRNAs are chemically synthesized using solid phase synthesis[10,11] and incorporate photocleavable residues containing a *o*-nitrobenzyl groups (Supplementary Fig. 1a) at defined positions. This *o*-nitrobenzyl group undergoes cleavage in response to UV light (Supplementary Fig. 1b), leaving a single phosphate group on the RNA fragment[12]. To develop a universal system, we tested a variety of sgRNA molecules, in which nucleotides at various positions[13] along the backbone have been replaced with photo-cleavable residues. (Supplementary Fig. 1c) Upon exposure to broad-spectrum light (80 mW cm$^{-2}$, Supplementary Fig. 1d), sgRNAs demonstrated fragmentation when analyzed on a fragment analyzer (Supplementary Fig. 1e). However, RNP complexes formed with some of these sgRNAs failed to initiate editing when delivered into cells (Supplementary Fig. 1f), potentially due to steric hindrance with Cas9 protein[14]. We identified two replacement sites that allowed sgRNAs to retain efficiency (positions 57 and 74, where position 1 is the 5′ end of the sgRNA, Supplementary Fig. 1c, g) and used these sites to create dual-breakage sgRNAs (DBsgRNA). DBsgRNAs were created as a fail-safe mechanism as cleavage at either site removes editing activity and increases the probability that at least one site is cleaved upon illumination. We found that these guides underwent fragmentation when irradiated with UV light as analyzed by electrospray ionization (ESI) mass spectrometry (Fig. 1b). Molecular weight of observed fragments corresponded closely to split sgRNAs at position 57 (fragments of 18 kDa and 14 kDa), sgRNAs split at position 74 (fragments of 8 kDa and 24 kDa), and split at both positions (fragments of 18 kDa, 8 kDa, and 5 kDa). DBsgRNAs also maintained in vitro editing activity comparable to standard sgRNAs when untreated and intact (Fig. 1c). This was determined by amplifying DNA containing the DNMT1 target sequence (Supplementary Table 1) and mixing with RNPs. We investigated sgRNA activity by assessing target DNA cleavage with a fragment analyzer. Importantly, when DBsgRNAs were synthesized as fragments and mixed with SpCas9, they did not exhibit any cleavage activity when assessed by the same assay (Fig. 1c).

We next demonstrated that DBsgRNAs are cleaved within cells upon illumination. Two hours following transfection of RNPs formed with standard sgRNAs or DBsgRNAs, cells were split into two populations, one being illuminated, and one kept in the dark to form paired experimental replicates. Both populations were allowed to recover for an additional 2 h, after which RNA was harvested. Using digital droplet PCR (ddPCR), we found that DBsgRNAs exposed to light exhibit a significant decrease in abundance in full-length DBsgRNA when compared to the paired population kept in the dark (Fig. 1d and Supplementary Table 3). This change was not observed using standard sgRNAs.

**CRISPRoff modulates genome editing events in human cells.** After determining DBsgRNAs can be effectively cleaved, we next tested the ability of CRISPRoff to modulate genome editing events within human cells. Due to the presence of potentially damaging UVA and UVB wavelengths present within our light source (Supplementary Fig. 1d), we first demonstrated that using a 345 nm long-pass filter did not significantly affect the viability of transfected HEK293 or U2OS cells (Supplementary Fig. 2a) and was used in all subsequent experiments.

Four hours following delivery of RNPs formed with DBsgRNAs targeting *DNMT1* into HEK293 cells, samples were illuminated for up to 60 s (Supplementary Fig. 2b) and allowed to recover for an additional 44 h. After harvesting the genomic DNA of these cells and analyzing amplified genomic target regions using Inference of CRISPR Edits (ICE)[15] we found that the degree of editing was significantly reduced in light-exposed samples (Fig. 1e). As a control, we also transfected HEK293s with standard sgRNAs and illuminated samples following the same protocol. Editing in illuminated sgRNA RNP populations was not significantly different than paired populations left in the dark. The similarity in overall editing efficiency following illumination of standard sgRNA RNPs suggests that DBsgRNAs were effectively cleaved within cells and no longer functional.

We further observed that illumination four hours post transfection retained a small portion of editing events, presumably from DSBs formed prior to external stimulus, including those that had not been repaired[16]. We reasoned that editing levels within populations may be titratable by modulating when the post-transfection timepoint at which samples are irradiated. To test this, we transfected HEK293s with DBsgRNA targeting *DNMT1* and illuminated a distinct cell sample, one time each, every two hours for two days. After 48 h, genomic DNA was isolated from all samples and analyzed for insertion/deletion mutations (indels). In alignment with our prediction, we were able to fine tune the level of gene editing within a population using DBsgRNAs (Fig. 1f).

**CRISPRoff is effective across cell lines and gene targets.** To test the universal effectiveness of the CRISPRoff system, we created a panel of standard sgRNAs and DBsgRNAs targeting a variety of chromosomes and local genomic contexts (Supplementary Table 1). Across all targets, all but two (CAMK1_sg2 and STK3_sg2) intact DBsgRNAs formed DSBs at a similar frequency as standard sgRNAs ($p < 0.05$, multiple independent t-test with FDR correction) (Fig. 2a–c) and generated a similar indel profile (Supplementary Fig. 2c). The majority of DBsgRNAs also showed a decrease in editing efficiency when illuminated four hours post transfection compared to cells from the same transfection that remained in the dark (Fig. 2a–c). Importantly, irradiation did not decrease editing efficiency of standard sgRNAs suggesting incorporation of photocleavable linkers was wholly responsible for the decrease in efficiency (Fig. 2a–c, right). We also observed that some targets were inactivated to a lesser degree than others, and hypothesized that the decrease in efficiency could be based on the individual editing kinetics at each site[17].

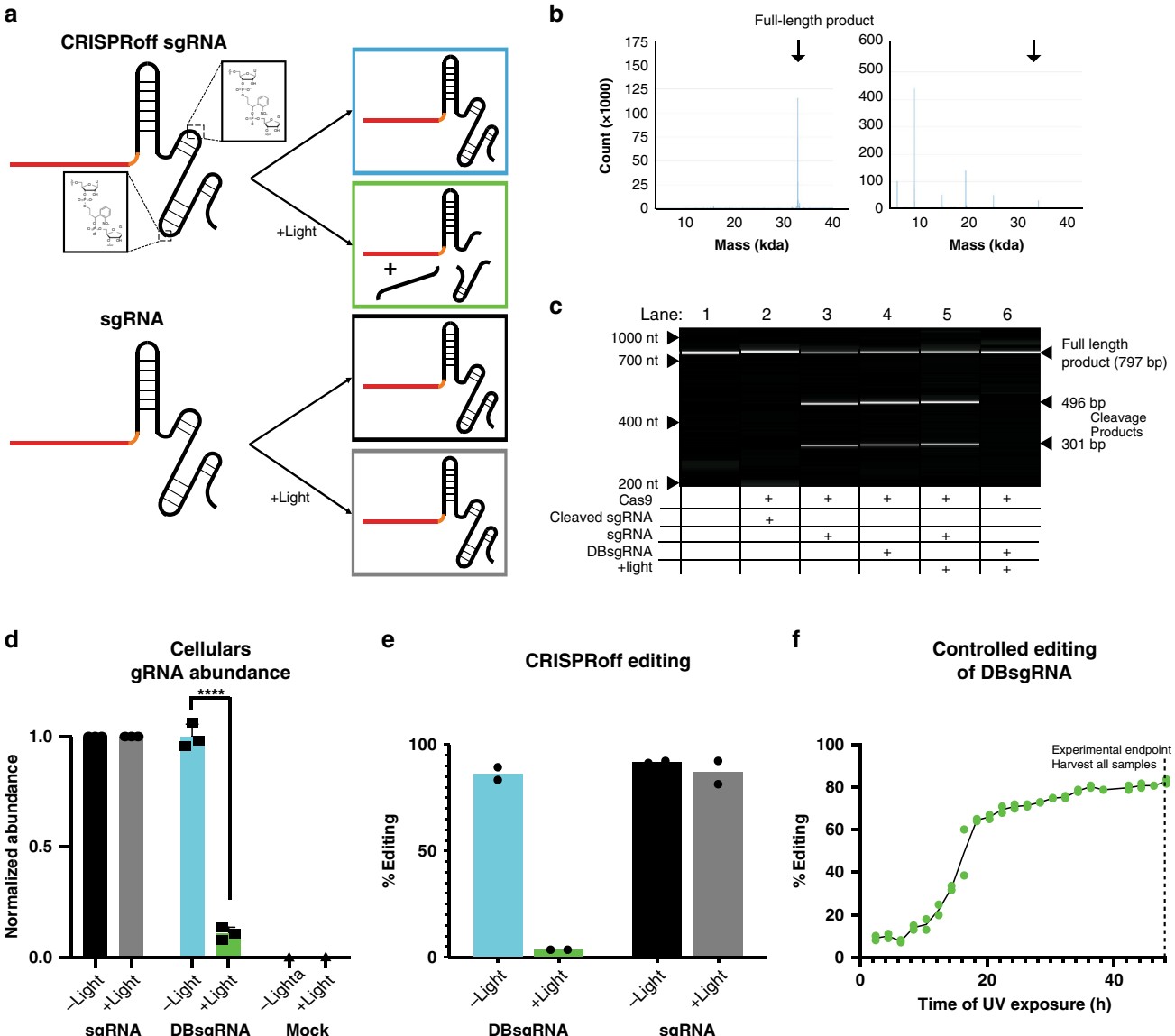

**Fig. 1 Characterization of CRISPRoff sgRNAs. a** Schematic diagram of CRIPSRoff dual-breakage (DB) sgRNAs. Nucleotides at positions 57 and 74 are replaced with residues that are cleaved in response to light. In the dark DBsgRNAs (blue) perform like standard sgRNAs (black). Upon exposure to light, the DBsgRNAs fragment (green) and are no longer functional. **b** Electrospray ionization (ESI) analysis of sgRNAs. Left: DBsgRNA before exposure to light. Product is full-length DBsgRNA. Right: DBsgRNA after exposure to light. Amount of full-length DBsgRNA is significantly reduced and observed products match predicted weights based on cleavage locations. **c** DNA fragment analysis showing in vitro cleavage of DNA induced by RNPs formed with sgRNA or DBsgRNAs. When synthesized as individual parts (lane 2), DBsgRNAs are unable to support substrate cutting. Similarly, after exposure to light DBsgRNAs fail to support target cleavage (lane 6). In all cases, RNPs formed with standard sgRNAs are able to cleave substrate (lane 3, 5). **d** Amount of sgRNA in cells analyzed by ddPCR with or without light exposure. Level of standard sgRNAs did not decrease in response to light. In contrast, DBsgRNAs were significantly depleted from cells following irradiation. Mock transfections did not detect sgRNA in either condition, indicating primers were specific to the target. ($n = 3$ experimental replicates, data is presented as mean ±1 SD, ****$p = 0.0003$, Student's two-tailed $t$-test with Bonferroni correction). **e** CRISPRoff editing in HEK293 cells at the *DNMT1* locus, DBsgRNAs formed indels at rates similar to standard sgRNAs in dark conditions. When exposed to light, DBsgRNAs were no longer able to induce indels ($n = 2$ paired experimental replicates, data is presented as mean). **f** Editing time course of DBsgRNAs targeting *DNMT1*. Distinct cell samples were exposed to light once each, with 2-h intervals between samples and all cells were harvested at 48 h for genetic analysis ($n = 2$ technical replicates). CRISPRoff allowed for titratable levels of genome editing within populations. Source data are provided as a Source data file.

To test this hypothesis, we identified one sgRNA, FANCF, that does not appear to be inactivated. To confirm that editing at this site can be controlled using CRISPRoff, we ran a high-resolution test of genome editing events where cells transfected with either standard or DBsgRNA RNPs were illuminated 15 min post transfection and harvested 15, 30, 60, 90, 120 min as well as longer time points at 4 and 24 h post transfection. In line with our

hypothesis, illumination of DBsgRNAs 15 min post transfection completely ablated editing at this site (Supplementary Fig. 2d). Interestingly, at four hours post transfection we observed that nearly 50% of alleles in the standard sgRNA transfection had already experienced a DSB that was repaired through NHEJ, confirming our hypothesis that the editing kinetics at this site is very fast.

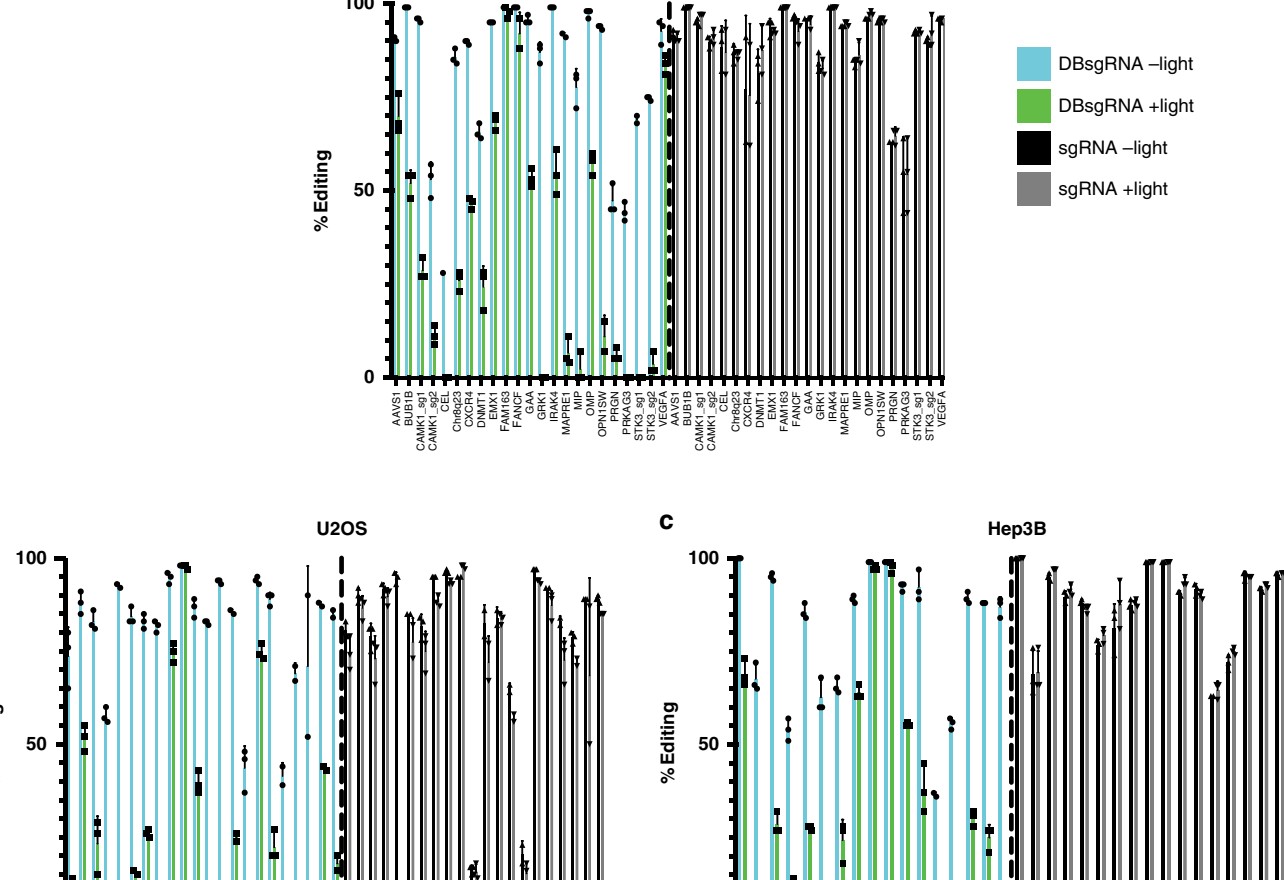

**Fig. 2 CRISPRoff sgRNAs are effective across cell lines and genomic targets. a–c** Panel of DBsgRNAs compared to standard sgRNAs in **a** HEK293 **b** U2OS, and **c** Hep3B cells. When left unstimulated, DBsgRNAs (blue, left) formed indels at rates similar to standard sgRNAs (black, right). In the cells exposed to light 4 h post transfection, editing was ablated for DBsgRNAs (green) but was unchanged using standard sgRNAs (gray) ($n = 3$ paired experimental replicates, data are presented as mean ±1 SD). Source data are provided as a Source data file.

**CRISPRoff can optimize on:off-target editing events**. Within our panel, we included an sgRNA known to be cytotoxic due to having an off-target site in an essential gene. Interestingly, when DBsgRNAs of this guide was used in conjunction with irradiation, a greater proportion of cells survived (Supplementary Fig. 2e), potentially due to an increase in the ratio of on:off-target events, while maintaining editing efficiency. With this in mind, we created sgRNAs that had significant levels of off-target editing at one or two sites within the genome (Supplementary Table 4). Based on previous studies, editing at off-target sites may be slower than editing at the on-target sites[18,19], and depend on RNP concentration within the cell[20,21]. We rationalized that we may be able to maximize the ratio between on:off-target editing (Supplementary Fig. 3a) by illuminating DBsgRNAs at an optimal time point post transfection. We transfected independent pools of cells with 7 unique sgRNAs and exposed the pools to light at 4, 8, 16, 24, or 48 h post transfection. We also harvested genomic DNA from each of these pools at the indicated time point to form a longitudinal editing curve. Following illumination, the degree of editing at many off-target sites plateaued, demonstrating that inactivating DBsgRNAs slowed down off-target editing (Supplementary Fig. 3b). By illuminating DBsgRNAs at discrete times

post transfection we found we were able to modulate and maximize the on:off-target cutting ratios (Fig. 3a).

**CRISPRoff enables precise spatial patterning**. One of the major advantages of using optical as opposed to chemical stimulus is the ability to obtain precise spatial control. This ability enables researchers to study complicated signaling effects such as paracrine vs juxtracrine signaling within a single well or better understand the role of specific genes during differentiation or organoid formation. Further uses in vivo could also help understand the effects of gene knockout in a developing embryo at a 2- or 4-cell state by laser illumination[22]. As a proof-of-concept, we obtained a GFP-expressing cell line[23] and designed sgRNAs to create GFP knock-out phenotypes. We used a standard inverted fluorescent micro-scope which could illuminate a single well at a time. This fluorescent microscope setup contained a 385 nm LED commonly used for illumination, and that is right on the edge of the reactivity of the PC linker in DBsgRNAs (Supplementary Fig. 1b). Using this setup, our DBsgRNA protocol continued to modulate indel formation in human cells (Fig. 3b). As a final study, we created a thin-film mask with transparent patterns with a precision of 8 µm (Fig. 3c). By

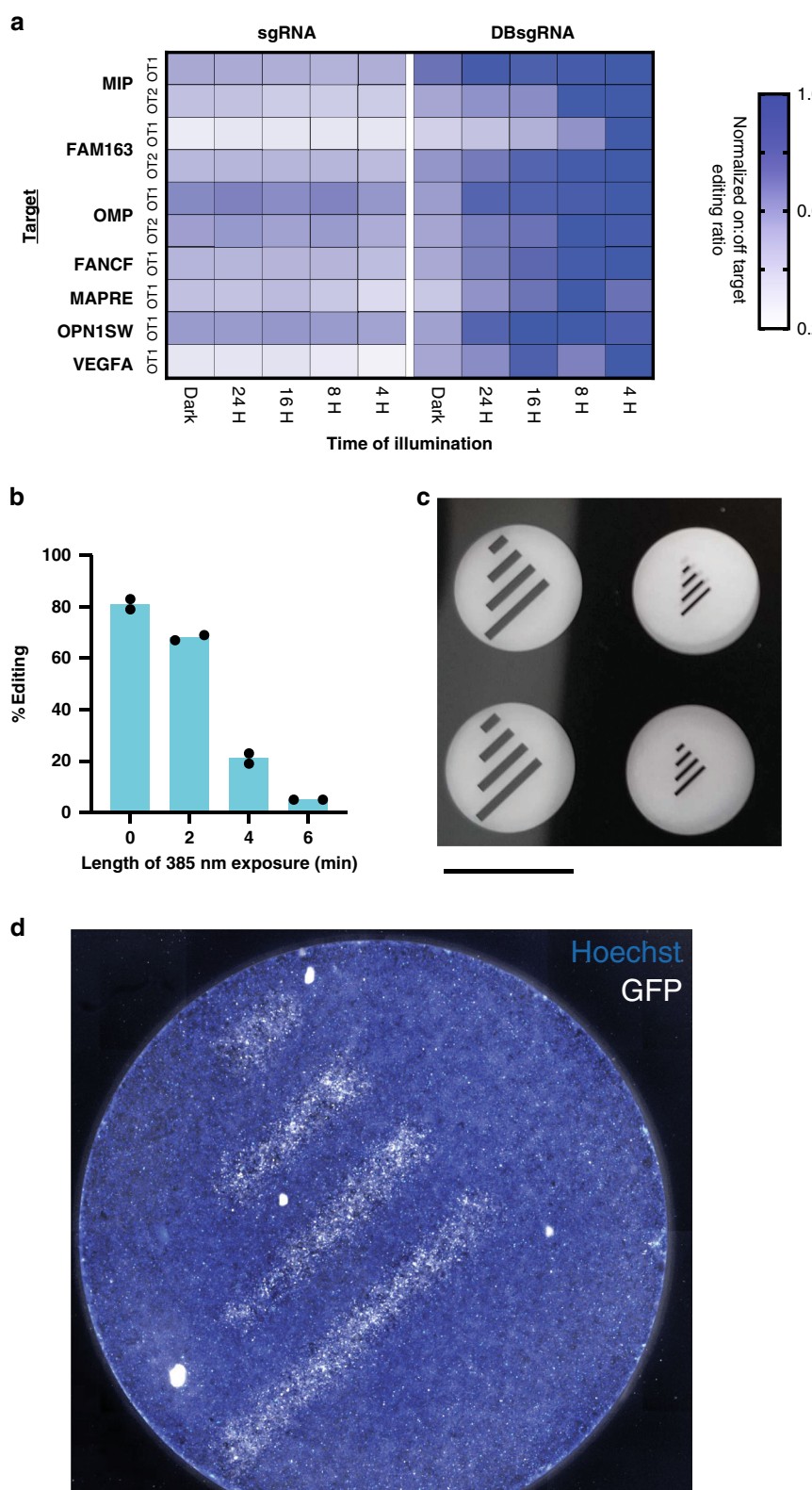

selectively masking the bottom of the well, we created distinct spatial patterns by knocking out GFP in defined regions. (Fig. 3d).

## Discussion

Taken together, CRISPRoff allows for tight control of editing from both a spatial and temporal perspective, expanding the toolbox of optogenetic gene editing[24–26]. We have successfully demonstrated this technology in multiple human cell lines across multiple genomic loci and expect this technology to be turn-key ready with any CRIPSR-based application currently using synthetic or to replace in vitro transcribed sgRNAs. While, at the moment, CRISPRoff is limited in in vivo applications, due to the

**Fig. 3 Applications of CRISPRoff sgRNAs. a** Ratio of on:off-target editing at known off-target sites (OT1 and OT2) for seven sgRNA targets. Ratios were normalized to the maximum on:off ratio as 1 (dark blue) to a minimum of 0.2 (white). Absolute values can be seen in Supplementary Fig. 3. Samples were exposed to light at the indicated time point and harvested after 48 h. DBsgRNAs had a higher on:off-target editing ratio when inactivated at earlier time points. This advantage decreased as editing is allowed to continue. For all targets, standard sgRNAs had the same on:off-target editing regardless of light exposure. **b** DBsgRNAs are inactivated using LED light from a standard epifluorescent microscope using a 385 nm LED. Full wells were irradiated for the indicated amount of time and analyzed by ICE. Exposure of 6 min ablated editing activity ($n = 2$ technical replicates, data is presented as mean). **c** Thin-film mask for spatial patterning using DBsgRNAs. Clear areas are transparent, allowing light to pass through and deactivate DBsgRNAs. Dark areas are opaque and editing continues unimpeded (scale bar: 6.5 mm). **d** Spatial pattering of gene editing using DBsgRNAs targeting GFP. Cells were irradiated through a negative mask. Editing was terminated in illuminated areas, preserving GFP fluorescence (scale bar: 1 mm). Source data are provided as a Source data file.

low penetrance of UV light through tissues, we are excited by the possibility of further chemistries extending the range of photo-cleavable molecules such as with two-photon cleavage systems[27,28]. Further, because CRISPRoff makes modifications to the backbone of the sgRNA, it can be compatible with other technologies, such as sgRNA modifications to activate gene editing[25], or Cas9 modifications to enhance on-target specificity[29,30]. We anticipate that the CRISPRoff system will be a valuable tool for both in vitro and in vivo control of CRISPR technologies.

## Methods

**RNA synthesis**. RNA oligonucleotides were synthesized on Synthego solid-phase synthesis platform, using CPG solid support containing a universal linker. 5-Benzylthio-1H-tetrazole (BTT, 0.25 M solution in acetonitrile) was used for coupling, (3-((Dimethylamino-methylidene)amino)-3H-1,2,4-dithiazole-3-thione (DDTT, 0.1 M solution in pyridine) was used for thiolation, dichloroacetic acid (DCA, 3% solution in toluene) for used for detritylation. After synthesis, oligonucleotides were subject to series of deprotection steps, followed by purification by solid phase extraction (SPE). Purified oligonucleotides were analyzed by ESI-MS. All materials for RNA synthesis were obtained from either ChemGenes or Thermo Fisher Scientific. CRISPRoff sgRNAs were made with PC Linker phosphoramidite, which was obtained from Glen Research (10–4920).

**Cell culture**. Human embryonic kidney cells (HEK293) and Hep3B were maintained between passage 5–20 in Advanced Modified Eagles Medium (Life Technologies) and 10% v/v FBS. Cells were passaged biweekly at a 1:8 ratio with TrypLE (Life Technologies).

U2OS cells were maintained between passage 5–15 in RPMI 1640 supplemented with 10% v/v FBS. Cells were passaged weekly at a 1:4 ratio with TrypLE. All cells were obtained from ATCC and maintained at 37 °C and 5% $CO_2$.

**Electrospray ionization**. RNA samples in TE buffer (3 uM) were analyzed by mass spectrometry (Agilent 1290 Infinity II liquid chromatography system (LC) coupled with Agilent 6530B Q-TOF mass spectrometer (MS)) in a negative ion polarity mode. LC is performed with gradient elution (buffer A: 50 mM HFIP; 15 mM Hexylamine 2% MeOH; buffer B: MeOH, 0.75 mL/min, 2–95% B in 1.05 min) on an Acquity UPLC BEH C18 VanGuard Pre-column (1.7 um, 2.1 × 5 mm). Electrospray ionization performed with a dual ESI source (gas temp 325 °C, drying gas 12 L/min, nebulizer 40 psi, Vcap 4 kV, fragmentor 250, skimmer 65). Data acquired in 100–3200 m/z range and deconvoluted in 4000–40000 m/z range.

**Fragment analysis**. Fragment analysis was done using a 5200 Fragment Analyzer System (Agilent) according to manufacturer protocols. DNA analysis was done using DNA small fragment kit (Agilent DNF-476) while RNA was analyzed using the small RNA kit (Agilent DNF-470).

**RNP formation and delivery**. 10 pmol *Streptococcus Pyogenes* [SV40 NLS]-[Sp. Cas9]-[SV40 NLS] protein (Aldevron Cat. #9212) was combined with 30 pmol synthetic sgRNAs (Synthego) in 20 μL total volume and allowed to complex for 10 min. During this incubation, cells were harvested and counted. To the RNP solution 5 μL of cell solution at a concentration of $4 × 10^4$ cells/μL was added and gently mixed.

Cell+RNP solution was transfected using the 4D-Nucleofector system (Lonza) in the 20 μL format. HEK293 transfections were conducted in SF buffer using protocol CM-130. U2OS and Hep3b transfections were conducted in SE buffer using protocol CM-104 and CM-130, respectively. Following transfection, cells were recovered in culture media and plated into 96-well plates. To create paired replicates, transfections were split into two pools. One that received light treatment while the other remained in the dark.

**DBsgRNA inactivation**. CRISPRoff inactivation was performed using a Sunray 600 UV Flood Lamp (Uvitron International). 345 nm, 6.5″ × 6.5″ colored glass alternative longpass filters were obtained from Newport.com and mounted using custom 3D-printed containers.

Inactivation using an upright microscope was performed using a Zeiss Axios Observer with a Colibri 7 Flexible Light Source and 385 nm LED. Imaging was preformed using a 4× objective lens.

**Genomic analysis**. Genomic DNA was isolated using DNA QuickExtract (Lucigen) following manufacturer protocol. After harvesting, extract solution was incubated at 65 °C for 15 min, 68 °C for 15 min followed by 98 °C for 10 min. Genomic PCR was performed using AmpliTaq Gold 360 Master Mix (Thermo Fischer) using primer sequences found in Supplementary Table 2. Following Sanger sequencing, presence of indels was analyzed via ICE (ice.synthego.com). Raw traces are also available at Zenodo 4009447.

**Digital droplet PCR**. Cellular RNA was extracted using RNA QuickExtract (Lucigen) without DNase. RNA was quantified using RiboGreen (Thermo Fisher) and normalized. Total RNA was reverse transcribed using iScript Advanced cDNA Synthesis Kit (BioRad) with 0.4 μM reverse primer for transcription. Reverse transcription product was amplified using 2x EvaGreen ddPCR Mastermix and thermal cycled at 95 °C for 3 min followed by 40 cycles of 95 °C for 30 s and 52.4 °C for 1 min. Signal was then stabilized at 4 °C for 5 min followed by inactivation at 90 °C for 5 min. Droplets were then read by a QX200 Droplet Digital PCR System and analyzed with QuantaSoft V1.7 (BioRad).

**Statistics**. All error bars are shown as ±1 SD. $p$ values were computed using Student's two-tailed $t$ test or one-way ANOVA and deemed significant at $α < 0.05$. Data was analyzed using Prism 8.

**Reporting summary**. Further information on research design is available in the Nature Research Reporting Summary linked to this article.

## Data availability
Sanger sequencing data supporting this work are available at Zenodo with accession code 4009447. All additional data is also available upon reasonable request. Source data are provided with this paper.

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

## Acknowledgements
We would like to thank Danielle Glick for help in designing and developing ddPCR assays and David Conant and Eric Zheng for careful reading of the manuscript. We further thank all members of Synthego for their support in this research.

## Author contributions
J.C-S, R.K., J.W., T.M. conceived the study, J.C-S., A.K., and S.J., designed and conducted experiments. N.R. performed bioinformatic analysis. J.C-S. wrote the paper with input from all authors.

## Competing interests
All authors declare a potential conflict of interest as employees or stockholders of Synthego Corporation. J.C-S, R.K., A.K., J.W., and T.M. have authored patent application PCT/US2020/015127 regarding CRISPRoff sgRNAs which has been filed by Synthego.
