## [Peer Review File · Nature Communications]

Reviewers' Comments:

Reviewer #1:

Remarks to the Author:

The manuscript by Carlson-Steevermer et al. describes work to create a system for genome editing that is responsive to light. Specifically, the authors have incorporated photocleavable residues into the backbone of the single-guide RNA, generating a targeted endonuclease that loses genome editing activity following exposure to UV light. The resulting enzyme indeed appears to be photoreactive, as a number of assays demonstrate. In particular, they present a strategy to kinetically control the ratio of on-target and off-target editing through the timing of exposure to UV light. This system has clear value, and the results seem encouraging. However, the work is ultimately impossible to assess because the reagents being used have not been described, including the modified guide RNA molecule that is central to the work.

This manuscript is not currently suitable for publication because it lacks adequate detail. In particular, no scientist would have any hope of replicating this work because the materials/methods do not describe the reagents being used. This is especially problematic because the central innovation relates to a novel, chemically-modified single-guide RNA (sgRNA) molecule. Because of this deficit, I recommend revision of the manuscript before it can be reconsidered for publication. This is exciting work of much utility, and I look forward to reading an updated version.

Essential recommended changes:

Perform a careful, critical reading of the manuscript to ensure that sufficient information is present to allow a skilled scientist (or team thereof) to replicate the work you performed. Such replication would currently be impossible because the design of the engineered sgRNA is not described in detail, and the manner by which it was produced is not described at all. The RNA is said to be modified at positions 57 and 74, but the two stars marking these sites in the sgRNA cartoon (Fig. 1a) don't clearly convey the positions relative to the important structural features of the sgRNA. The authors cite Briner et al., and it would be useful to annotate a cartoon like the one used by Briner & colleagues where the two modified sites are (especially relative to the nexus feature).

Beyond this central deficit, there are other topics that require additional detail. The source and nature of the Cas9 protein is poorly described. The authors claim to have sourced "NLS-Cas9-NLS" protein from Aldevron, but this term does not appear to refer to a specific product. Either specify a product number/name, or describe the protein in detail, or – ideally – do both. The manuscript does not even note the organism of origin for the Cas9 being used (*S. pyogenes*, *S. aureus*, something else?).

The description of nucleofection protocol should include which program/setting was used for nucleofection.

Additional suggestions to help improve the manuscript:

In Fig. 1a (or a new figure in the supplement), it would be helpful to show where/how the ddPCR primers map onto the sgRNA structure. Currently, the information in Supplementary Table 3 is essentially useless.

In Fig. 1f, it is distracting and confusing that error bars are only sometimes present. Since only two replicates were performed, consider omitting the error bars and instead show each data point, along with a line for the average value. The X-axis label is confusing, and perhaps misleading. Instead of "CRISPR activity", consider a clear, precise label along the lines of "time of UV exposure". It will be important to make this clear, since at first glance the plot (as submitted) seems to suggest an increase in activity over time. I may be misunderstanding the experiment, however. The materials & methods section says "unique cell samples were exposed to light every two hours", which could be misleading. It sounds as if any given sample was exposed to light

multiple times. My understanding is that distinct cell samples were each exposed to light once each, and that the time at which exposure was performed was chosen from a "schedule" of every two hours (forming the X-axis of the plot in question). Also, if the blue dot at 48 h (?) means something, please tell us what that means. I presume this was a no-UV sample.

In Fig. 2a/b/c, it would be helpful to put a legend within the figure. It's hard to interpret at first glance. This could be placed at the top of 2a, and wouldn't need to take up too much space.

Supp. Table 4: it would be convenient to see the corresponding on-target sites here (even though it's technically redundant with Supp. Table 1).

The following comments correspond to the main text, with a page/line format of P#L##

P1L21 It is inaccurate to claim that RNPs will form DSBs "everywhere" that the sgRNA binds within the genome.

P2/L33 Remove comma, giving "However, RNP complexes formed".

P2L37 Remove "total" since there is no evidence in support of the idea that the DBsgRNAs underwent "total fragmentation".

P2L41 Consider replacing "during these same assays" to "when assessed by the same assays".

P2L44 As above, remove "completely" from "completely cleaved" if there is no evidence presented to support this.

P2L46 Update "one for illumination" to "one being illuminated"

P3L47 Add comma here: "allowed to recover for 2 hours, after which"

P3L54 I think it could be useful to specify that electroporation/nucleofection was performed; many readers might assume something else if they only see the term "transfection". It describes "transfection of DBsgRNAs" but I believe it would be more accurate to refer to delivery of "RNPs formed using DBsgRNAs". It could be useful to establish the concept of a "DBsgRNA RNP", allowing use of this shorthand.

P4L79-80 "Each" should not be used to describe the responsiveness of the DBsgRNA RNPs to light, since some of them were not responsive. Decide what your threshold is and be descriptive about which DBsgRNA RNPs did or did not have their activity respond to UV exposure. Omit "We determined that" and start the sentence with "We suspect that the decrease in..." and adjust the subsequent sentence to match. Unless I am misunderstanding, you are proposing a potential/likely explanation for the lack of responsiveness in some DBsgRNA RNPs. This is welcome, but it should not be presented as anything you determined.

P4L90 It is claimed that "cells survived" as if cell survival is all-or-nothing. I don't think that's what your data suggest. Please carefully and thoughtfully revisit this section. The cell survival does not appear to be binary, and frankly the images in Fig. S2d are a bit difficult to interpret. Repeating this experiment with staining for dead cells might result in a more readily interpretable dataset.

P4L92 Point to Supp. Table 4 here.

P5L95-96 Consider updating to "maximization may be facilitated by a lag period"

P5111 Be thoughtful with your claims: this approach is not compatible with "any CRISPR-based application". Any application relying on viral vector delivery would not be compatible.

P5115 I am wary about using the term "CRISPR" to refer to "CRISPR-mediated genome editing". These are two distinct ideas.

Reviewer #2:

Remarks to the Author:

In their manuscript CRISPRoff: A spatio-temporally tunable CRISPR system, Carlson-Stevermer et al. describe a light-dependent CRISPR-Cas9 off-switch based on photocleavable sgRNAs. The authors show that RNP complexes comprising a photocleavable sgRNAs and Cas9 enable UV light-dependent genome editing in mammalian cells. The authors also investigate possibilities to reduce off-target editing by blocking Cas9 at specific time points post-delivery and showcase the potential of their system for spatially confined genome editing.

Albeit this study is of considerable interest and I acknowledge the novelty of light-inactivated sgRNAs, I do not see that the manuscript lives up to the standard expected for a Nature Communications paper in its present form (see my points below). That said, the manuscript could be highly strengthened, if the authors would expand their (experimental!) analysis with respect to the applications of their technology and limitations of thereof (see major points 1 and 2, respectively). Finally, and very importantly: the manuscript completely lacks any details with respect to the design of the light-deactivated sgRNA, which arguably is the single most-important advance presented in this manuscript. What is the underlying (photo)chemistry? How were the photosensitive sgRNAs synthesized? What considerations underlie the choosing of sgRNA sites used to incorporate the photosensitive moiety? This information is absolutely essential to enable others to build upon this technology as well as to judge the data provided in this manuscript and therefore must be included.

Major points:

1) The authors limited application of CRISPRoff to Cas9 and specifically to genome editing. However, as the authors "expect this technology to be compatible with any CRISPR-based application", I kindly ask the authors to please exemplify this versatility in additional experiments. First, the authors should show that their technology is compatible with dCas9, e.g. by performing experiments based on CRISPRi or CRISPRa. Such experiments would highly strengthen the manuscript. Also, the authors should strongly consider transferring their technology to a second Cas9 orthologue or, even better, to Cas12 or Cas13 RNA guides, which, again, would highly strengthen their manuscript and support the claim made by the authors.

2) With respect to point 2: Looking at the data presented in Fig. 2a-c, the CRISPRoff technology seems to work well only for a subset of target sites/sgRNAs. In a number of cases, considerable and sometimes even very strong editing is observed even in the illuminated case, i.e. when CRISPR should be blocked (e.g. FAM163 and FANCF). The authors reason that this would be due to the editing kinetics and state that "[...] the timepoint of irradiation must be optimized for individual targets when using the CRISPRoff system". While the authors provide a fair hypothesis, they should investigate whether it was indeed true, since this is of particular importance for users. Thus, I ask the authors to please perform additional experiments for some of the sgRNAs in Fig. 2a-c that did show strong editing in the presence of light and demonstrate that an earlier time point of irradiation resolves the problem of unintended editing.

3) Experimental replicates and error bars: Most of the presented data, even in the two main figures, are either technical replicates or "paired experimental replicates". Are the latter independent experiments or not (I guess not)? I ask the authors please independently reproduce all of their main experiments by performing experiments independently and on different days at least three times to ensure reproducibility of their findings. Also, what do the error bars indicate? SD? Add this information to all figure legends, where applicable.

4) Fig 2d: Please show absolute ON- and OFF-target editing frequencies, either in Fig. 2d or as additional Supplementary Figure (similar to Fig. S2e). I ask this, since there could be cases, in which the reported, normalized On:Off target ratio is high, but the overall editing efficacy could – unintentionally – be very low.

Minor points:

1) Lines 31/32: “Upon exposure to unfiltered UV light”: What do the authors mean by “unfiltered”. Which wavelength(s) did they use specifically?

2) Lines 44-48: The experimental timing is not clear to me. At which time post RNP delivery were cells illuminated and for how long were they incubated post illumination, i.e. before lysing cells and running the assay?

3) Line 59/60: “Editing in these samples were not significantly different than samples left in the dark”. Could be misunderstood, as it is not necessarily clear what the “samples” refer to in each case. Clarify, e.g. by stating “Editing in the control samples were not significantly different than in the DBsgRNA samples left in the dark” or similar.

4) Lines 61-63: The authors state “Further optimization of CRISPRoff in both HEK293s and U2OS cell lines showed that using a 355 nm longpass filter supported deactivation while maintaining cell viability (Fig. S2b)”. However, Fig. S2b only shows maintained viability, but not sgRNA deactivation, i.e. parts of the claim are not supported by the referenced data.

5) Lines 83/84: “however, this site is nearly 100% edited within four hours of transfection” Where is the data supporting this claim?

6) The authors state (lines 79-81): “Each DBsgRNA also showed a decrease in editing efficiency when illuminated four hours post transfection (Fig. 2A-C).” In several cases there is practically no difference between the light and dark samples, see data on FAM163 and FANCF. Please revise statement.

7) Lines 94-96: “By analyzing genomic DNA from cells at various time points post-transfection, we suggest this maximization may be caused by a lag period when forming off-target indels (Fig. S2e).” I am not sure that I get the argument the authors try to make here. Do the authors wish to say that off-target edits were reduced because they follow a slower kinetics and thus need more time to occur as compared to on-target edits? Please clarify

8) The information which light intensities were used to trigger sgRNA cleavage needs to be added to the methods/figure legends.

9) Fig. 1d: Plotting data without prior normalization would be more informative, since this would allow resolving potential differences between sgRNA and DBsgRNA delivery efficacies

10) Fig. S2a: Why does editing in the sgRNA control samples decrease with prolonged UV exposure?

11) What does “Mod” refer to in Fig. S1? Also, I do not understand the origin of the many bands in Fig. S1A. Should there not be only two bands visible in the illuminated samples? The authors should consider providing a scheme of the different sgRNA designs and corresponding cleavage fragments, which would highly simplify interpretation of the bands in Fig. S1A.

Reviewer #3:

Remarks to the Author:

In this paper, Stevermer et. al. present a method for controllable CRISPR using a guide RNA with a photocleavable linker. Overall, the work is well done and of general interest but with overstated results and lacking detail on experimental methods. I think if these things are fixed then it could be publishable in Nature Communications. My main comments (below) are related to adding experimental details, improving the figures, and not overstating results.

Major comments:

- Figure 2 a/b/c is VERY difficult to make sense of the way the left and right panels are plotted. I think you have to somehow merge left and right panels or think about a different way to plot your results that aren't so difficult to decipher. Also it would be useful if you can line up for the 3 cell lines the common targets (e.g. FANCF shifts right in 2c). Personal suggestion – I'd probably merge left and right into one with 4 bars at each target and remove the data points for clarity and include those in SI instead.
- Overstatements that need repair:
 - “However, each of these strategies requires a new, non-native component of the CRISPR system.” First, this doesn't seem entirely true, and second how does yours also not fall into this category?
 - “We next demonstrated that DBsgRNAs are completely cleaved within cells upon illumination” – not “complete” RNA cleavage.
 - “intact DBsgRNAs formed DSBs at a similar frequency as standard sgRNAs”. This statement is neither quantitative nor easily verifiable from your data. It feels like you're trying to sweep under the rug the many cases where this is not true.
 - “We determined that the decrease in efficiency is based on the individual editing kinetics at each site” – this appears to be an educated guess or a suspicion. You can either prove it or tone it down. Also as a counterexample EMX1 changes editing dramatically between cell lines when presumably the kinetics of editing probably doesn't.
 - “One of the major advantages of using optical opposed to chemical stimulus is the ability to obtain precise spatial control.” – This should have an explanation. I could see an advantage in tissues, but then you run into low UV transmission and potential tissue damage that will probably make this method not useful. You have a nice demo of spatial control, but you need to give examples of what it could be used for. I personally can't think of one for cells, given that you could always physically split them instead.
 - “We have successfully demonstrated this technology in multiple human cell lines across multiple genomic loci and expect this technology to be compatible with any CRISPR-based application. “ - you forgot the “m” on many
- Fragmentation
 - Report estimated band lengths in Figure S1A.
 - The full sequence and length of your sgRNAs are not reported anywhere
 - Missing details on synthesis/purchase of photocleavable RNA

- Why are there sometimes two and sometimes 3 fragments for a single cleavage location?
 - Figure S1 – Abbreviations are given without any explanation or context (“mod” “DNMT1” “FANCF” “VEGFA”)
 - Experimental details of the fragmentation assay and analysis are not provided.
- Figure 1
- Panel b - “minor products match predicted weights based on cleavage locations” is not at all obvious, or verifiable based on the information we’re given. It appears a ~33 kDa product (with a ~17 kDa unidentified sub-product) degrades into fragments of ~24, 18, 14, 8. Based on my crude calculations I’d expect fragments of ~19, 9, and 6 kDa. Please resolve this.
 - Also, visually the arrows are not descriptive on their own.
 - Panel c – This experiment is not described in (any) sufficient detail. After some time I get what is going on but it is missing 1) experimental protocol, 2) target dsDNA info, 3) sgRNA info (full length and fragment)
 - Panel d + e
 - Similar to c, target dsDNA and sgRNA sequences are not given.
 - something should be added to visually distinguish these two. d shows RNA cleavage and e shows % editing in cells. A short title for each or a cartoon could help.
 - Panel f – I don’t see how this shows controlled gene editing without a proper control. The x-axis doesn’t make sense – CRISPR Activity with units of hours?
- Illumination
- Is UV going to be a problem for passing through biological materials? Also UV is known to damage DNA, so maybe you can make some comments on those.
 - Why broad spectrum light in Figure S2a? I think you mean broad spectrum UV, but it is not explained. The wavelength ranges should be stated. What is the source?
 - Since there is no difference between 345 and 355 filters in S1b, how is the 355 filter an optimization?
 - Why do you use 385 nm in Figure 2? Choice is not obvious and not explained.

Minor comments:

- ICE – unabbreviate at first instance and give brief description.
- “indel” describe/unabbreviate at first use.
- S2c would be clearer if you used 1 panel with two lines for DBsgRNA and sgRNA
- I would not use $h\nu$ as a shorthand for light. First it only saves you 3 letters, second I don’t think everyone will get it. Also, you write “v” incorrectly for the Greek nu in many instances, further confusing it.
- Figure 2e is distracting (and the result is obvious from 2f) – I would personally replace it with some version of S2f to show why the pattern emerges.
- “OT1” and “OT2” abbreviations are not explained anywhere (Figure 2).

Reviewer #1 (Remarks to the Author):

The manuscript by Carlson-Stevermer et al. describes work to create a system for genome editing that is responsive to light. Specifically, the authors have incorporated photocleavable residues into the backbone of the single-guide RNA, generating a targeted endonuclease that loses genome editing activity following exposure to UV light. The resulting enzyme indeed appears to be photoreactive, as a number of assays demonstrate. In particular, they present a strategy to kinetically control the ratio of on-target and off-target editing through the timing of exposure to UV light. This system has clear value, and the results seem encouraging. However, the work is ultimately impossible to assess because the reagents being used have not been described, including the modified guide RNA molecule that is central to the work.

This manuscript is not currently suitable for publication because it lacks adequate detail. In particular, no scientist would have any hope of replicating this work because the materials/methods do not describe the reagents being used. This is especially problematic because the central innovation relates to a novel, chemically-modified single-guide RNA (sgRNA) molecule. Because of this deficit, I recommend revision of the manuscript before it can be reconsidered for publication. This is exciting work of much utility, and I look forward to reading an updated version.

We thank the reviewer for their careful reading and kind words regarding the capabilities of this new technology. We agree that the first draft had significant details missing. This has been substantially updated in the revised manuscript.

Essential recommended changes:

Perform a careful, critical reading of the manuscript to ensure that sufficient information is present to allow a skilled scientist (or team thereof) to replicate the work you performed. Such replication would currently be impossible because the design of the engineered sgRNA is not described in detail, and the manner by which it was produced is not described at all. The RNA is said to be modified at positions 57 and 74, but the two stars marking these sites in the sgRNA cartoon (Fig. 1a) don't clearly convey the positions relative to the important structural features of the sgRNA. The authors cite Briner et al., and it would be useful to annotate a cartoon like the one used by Briner & colleagues to communicate where the two modified sites are (especially relative to the nexus feature).

We agree with the reviewer about the inclusion of important details related to RNA synthesis. To this end we have updated the method section with relevant non-proprietary details related to our RNA synthesis. We have further included an updated diagram (Fig S1c.) that has exact location of the linker molecule and relation to other relevant structures.

Beyond this central deficit, there are other topics that require additional detail. The source and nature of the Cas9 protein is poorly described. The authors claim to have sourced "NLS-Cas9-NLS" protein from Aldevron, but this term does not appear to refer to a specific product. Either specify a product number/name, or describe the protein in detail, or – ideally – do both. The manuscript does not even note the organism of origin for the Cas9 being used (*S. pyogenes*, *S. aureus*, something else?).

The reviewers bring up an important point about distinguishing the specific Cas9 used. In these experiments we have not used any form of engineered variants. We have updated mentions of Cas9 to be *Sp.Cas9* and included available catalog numbers.

The description of nucleofection protocol should include which program/setting was used for nucleofection.

Methods have been updated to include the specific 4D-nucleofector protocol used for each cell line.

Additional suggestions to help improve the manuscript:

In Fig. 1a (or a new figure in the supplement), it would be helpful to show where/how the ddPCR primers map onto the sgRNA structure. Currently, the information in Supplementary Table 3 is essentially useless.

We believe the method we have developed to detect the presence of sgRNAs within the cell may offer value to the field at large, particularly within cell therapy. To this end, we have included a new supplementary figure (Fig S1c) to show precisely how the ddPCRs primers map on to the scaffold of the sgRNA.

In Fig. 1f, it is distracting and confusing that error bars are only sometimes present. Since only two replicates were performed, consider omitting the error bars and instead show each data point, along with a line for the average value. The X-axis label is confusing, and perhaps misleading. Instead of “CRISPR activity”, consider a clear, precise label along the lines of “time of UV exposure”. It will be important to make this clear, since at first glance the plot (as submitted) seems to suggest an increase in activity over time. I may be misunderstanding the experiment, however. The materials & methods section says “unique cell samples were exposed to light every two hours”, which could be misleading. It sounds as if any given sample was exposed to light multiple times. My understanding is that distinct cell samples were each exposed to light once each, and that the time at which exposure was performed was chosen from a “schedule” of every two hours (forming the X-axis of the plot in question). Also, if the blue dot at 48 h (?) means something, please tell us what that means. I presume this was a no-UV sample.

We have modified the figure to include both independent data points. In many cases the level of editing as assayed was the same in both data points. This caused the error bars to appear to be missing. We further thank the reviewer for their suggestion about wording of the legend and methods. We have reworked this language to bring more clarity to the experiment.

In Fig. 2a/b/c, it would be helpful to put a legend within the figure. It’s hard to interpret at first glance. This could be placed at the top of 2a, and wouldn’t need to take up too much space. A legend has been included in Figure 2.

Supp. Table 4: it would be convenient to see the corresponding on-target sites here (even though it’s technically redundant with Supp. Table 1).

On-target sites have been added for comparison in Supp Table 4.

The following comments correspond to the main text, with a page/line format of P#L##

P1L21 It is inaccurate to claim that RNPs will form DSBs “everywhere” that the sgRNA binds within the genome.

P2/L33 Remove comma, giving “However, RNP complexes formed”.

P2L37 Remove “total” since there is no evidence in support of the idea that the DBsgRNAs underwent “total fragmentation”.

P2L41 Consider replacing “during these same assays” to “when assessed by the same assays”.

P2L44 As above, remove “completely” from “completely cleaved” if there is no evidence presented to support this.

P2L46 Update “one for illumination” to “one being illuminated”

P3L47 Add comma here: “allowed to recover for 2 hours, after which”

P3L54 I think it could be useful to specify that electroporation/nucleofection was performed; many readers might assume something else if they only see the term “transfection”. It describes “transfection of DBsgRNAs” but I believe it would be more accurate to refer to delivery of “RNPs formed using DBsgRNAs”. It could be useful to establish the concept of a “DBsgRNA RNP”, allowing use of this shorthand.

P4L79-80 “Each” should not be used to describe the responsiveness of the DBsgRNA RNPs to light, since some of them were not responsive. Decide what your threshold is and be descriptive about which DBsgRNA RNPs did or did not have their activity respond to UV exposure. Omit “We determined that” and start the sentence with “We suspect that the decrease in...” and adjust the subsequent sentence to match. Unless I am misunderstanding, you are proposing a potential/likely explanation for the lack of responsiveness in some DBsgRNA RNPs. This is welcome, but it should not be presented as anything you determined.

P4L90 It is claimed that “cells survived” as if cell survival is all-or-nothing. I don’t think that’s what your data suggest. Please carefully and thoughtfully revisit this section. The cell survival does not appear to be binary, and frankly the images in Fig. S2d are a bit difficult to interpret. Repeating this experiment with staining for dead cells might result in a more readily interpretable dataset.

P4L92 Point to Supp. Table 4 here.

We have assimilated the changes proposed by the reviewer above into the manuscript.

P5L95-96 Consider updating to “maximization may be facilitated by a lag period”

We have updated the manuscript to include these improvements to readability and comprehension.

P5111 Be thoughtful with your claims: this approach is not compatible with “any CRISPR-based application”. Any application relying on viral vector delivery would not be compatible.

We have updated our considerations to include CRISPR-based using exogenous sgRNAs. Virally encoded Cas9 and a lipofected sgRNA would still be a compatible application.

P5115 I am wary about using the term “CRISPR” to refer to “CRISPR-mediated genome editing”. These are two distinct ideas.

The reviewer brings up an interesting point regarding nomenclature of the field of genome manipulation. While they are correct we only show CRISPR-mediated genome editing, we do, however, foresee applications in spaces such as CRISPRa, CRISPRi or in future iterations of CRISPRoff, Cas13-mediated RNA control.

Reviewer #2 (Remarks to the Author):

In their manuscript CRISPRoff: A spatio-temporally tunable CRISPR system, Carlson-Stevermer et al. describe a light-dependent CRISPR-Cas9 off-switch based on photocleavable sgRNAs. The authors show that RNP complexes comprising a photocleavable sgRNAs and Cas9 enable UV light-dependent genome editing in mammalian cells. The authors also investigate possibilities to reduce off-target editing by blocking Cas9 at specific time points post-delivery and showcase the potential of their system for spatially confined genome editing.

Albeit this study is of considerable interest and I acknowledge the novelty of light-inactivated sgRNAs, I do not see that the manuscript lives up to the standard expected for a Nature Communications paper in its present form (see my points below). That said, the manuscript could be highly strengthened, if the authors would expand their (experimental!) analysis with respect to the applications of their technology and limitations of thereof (see major points 1 and 2, respectively). Finally, and very importantly: the manuscript completely lacks any details with respect to the design of the light-deactivated sgRNA, which arguably is the single most-important advance presented in this manuscript. What is the underlying (photo)chemistry? How were the photosensitive sgRNAs synthesized? What considerations underlie the choosing of sgRNA sites used to incorporate the photosensitive moiety? This information is absolutely essential to enable others to build upon this technology as well as to judge the data provided in this manuscript and therefore must be included.

We thank the reviewer for their careful reading of the manuscript and acknowledgment of the utility of this technology. We agree that further experimental detail regarding the synthesis of CRISPRoff sgRNAs is necessary and have expanded this discussion throughout the manuscript as well as in the methods.

Major points:

1) The authors limited application of CRISPRoff to Cas9 and specifically to genome editing. However, as the authors “expect this technology to be compatible with any CRISPR-based application”, I kindly ask the authors to please exemplify this versatility in additional experiments. First, the authors should show that their technology is compatible with dCas9, e.g. by performing experiments based on CRISPRi or CRISPRa. Such experiments would highly strengthen the manuscript. Also, the authors should strongly consider transferring their technology to a second Cas9 orthologue or, even better, to Cas12 or Cas13 RNA guides, which, again, would highly strengthen their manuscript and support the claim made by the authors.

While we agree that these experiments proposed by the reviewer are interesting, we chose to demonstrate the effectiveness of the CRISPRoff system using Sp.Cas9 genome editing due to it being the largest use case of CRISPR technologies within the community.

One of the strengths of the CRISPRoff system is the ability to deliver as an RNP rather than creating engineered cell lines or delivering machinery via plasmid. To the best of our knowledge there are no commercially available proteins to conduct CRISPRi or CRISPRa experiments and these effects would be transient at best. If this application is desirable and integration of transgenes is allowed, a technology such as a split-Cas9 could be used (Zetsche 2015) to obtain temporal control.

While we believe the CRISPRoff system could be translatable to any CRISPR approach this work is out of scope for our current projects. To reflect this we have adapted the language to only explicitly state SpCas9 genome editing.

2) With respect to point 2: Looking at the data presented in Fig. 2a-c, the CRISPRoff technology seems to work well only for a subset of target sites/sgRNAs. In a number of cases, considerable and sometimes even very strong editing is observed even in the illuminated case, i.e. when

CRISPR should be blocked (e.g. FAM163 and FANCF). The authors reason that this would be due to the editing kinetics and state that “[...] the timepoint of irradiation must be optimized for individual targets when using the CRISPRoff system”. While the authors provide a fair hypothesis, they should investigate whether it was indeed true, since this is of particular importance for users. Thus, I ask the authors to please perform additional experiments for some of the sgRNAs in Fig. 2a-c that did show strong editing in the presence of light and demonstrate that an earlier time point of irradiation resolves the problem of unintended editing.

We thank the reviewer for their keen reading and observation of these points. Our intent was to provide a wide view of potential targets so that this technology could be widely adopted rather than highly optimizing on a few targets that may not be translatable. To this end we randomly chose sgRNAs from our database to use. We also made a conscious decisions to display all data, even if it might not clearly demonstrate our hypothesis at a single time point. We believe that this illuminates the differences of kinetics within CRISPR-based genome editing.

We have taken the reviewers suggestion and used one sgRNA with high editing (FANCF) and demonstrated that the DSB, and further, indel mutations can happen quickly and can be controlled by early exposure to light (Fig. S2e)

3) Experimental replicates and error bars: Most of the presented data, even in the two main figures, are either technical replicates or “paired experimental replicates”. Are the latter independent experiments or not (I guess not)? I ask the authors please independently reproduce all of their main experiments by performing experiments independently and on different days at least three times to ensure reproducibility of their findings. Also, what do the error bars indicate? SD? Add this information to all figure legends, where applicable.

To the reviewer’s point, the data for the figures was generated using paired experimental replicates. These are conducted by taking one shared pool of cells and splitting it into two groups (light or dark). In this manner we can be certain that the only difference between the samples is exposure to cleavage agent rather than transfection variability. Finally, we have demonstrated that our results hold across three different cell lines which were conducted independently. We have updated our results and analysis sections to ensure that this is stated more plainly.

We have updated the legend to explicitly state that errors bars are +/-1 standard deviation.

4) Fig 2d: Please show absolute ON- and OFF-target editing frequencies, either in Fig. 2d or as additional Supplementary Figure (similar to Fig. S2e). I ask this, since there could be cases, in which the reported, normalized On:Off target ratio is high, but the overall editing efficacy could – unintentionally - be very low.

We recognize the concern the reviewer has and have included a new supplementary figure 3 showing the absolute on and off target editing efficiency at each target at each time point. Normalized data was displayed in the main figure so as to not highlight outliers within the data set and minimize the variance between off-target sites.

Minor points:

1) Lines 31/32: “Upon exposure to unfiltered UV light”: What do the authors mean by “unfiltered”. Which wavelength(s) did they use specifically?

We thank the reviewer for bringing up this major point. The main source of light we used contained a spectrum of wavelengths. We have included this spectrum as well as that of the photocleavable linker in Figure S1 so that readers can see the overlap.

2) Lines 44-48: The experimental timing is not clear to me. At which time post RNP delivery

were cells illuminated and for how long were they incubated post illumination, i.e. before lysing cells and running the assay?

We have updated our discussion to better illustrate the exact experimental timing of these assays.

3) Line 59/60: "Editing in these samples were not significantly different than samples left in the dark". Could be misunderstood, as it is not necessarily clear what the "samples" refer to in each case. Clarify, e.g. by stating "Editing in the control samples were not significantly different than in the DBsgRNA samples left in the dark" or similar.

We have edited these lines to provide more clarity into the experimental conditions. We have incorporated this language into the manuscript.

4) Lines 61-63: The authors state "Further optimization of CRISPRoff in both HEK293s and U2OS cell lines showed that using a 355 nm longpass filter supported deactivation while maintaining cell viability (Fig. S2b)". However, Fig. S2b only shows maintained viability, but not sgRNA deactivation, i.e. parts of the claim are not supported by the referenced data.

We thank the reviewer for highlighting this omission. We have reviewed our data and changed the experimental layout to show both of these points (viability and deactivation).

5) Lines 83/84: "however, this site is nearly 100% edited within four hours of transfection" Where is the data supporting this claim?

We have added an additional supplementary figure to demonstrate this claim (Fig. S2e) and also demonstrate the ability to control editing at this site using earlier exposure. This point has been significantly expanded on within the manuscript.

6) The authors state (lines 79-81): "Each DBsgRNA also showed a decrease in editing efficiency when illuminated four hours post transfection (Fig. 2A-C)." In several cases there is practically no difference between the light and dark samples, see data on FAM163 and FANCF. Please revise statement.

We have added additional experiments to address this important point and have further expanded on the effects of editing kinetics at different sites.

7) Lines 94-96: "By analyzing genomic DNA from cells at various time points post-transfection, we suggest this maximization may be caused by a lag period when forming off-target indels (Fig. S2e)." I am not sure that I get the argument the authors try to make here. Do the authors wish to say that off-target edits were reduced because they follow a slower kinetics and thus need more time to occur as compared to on-target edits? Please clarify.

The reviewer has correctly intuited the argument we are trying to present. This hypothesis has been more clearly stated and supported with relevant literature.

8) The information which light intensities were used to trigger sgRNA cleavage needs to be added to the methods/figure legends.

We have added language to clarify light sources used in these experiments and have also added to the discussion about the variety of light sources that can be used. Further we have supplemented Fig. S1 with data showing the cleavable spectrum of CRISPRoff and the light sources used.

9) Fig. 1d: Plotting data without prior normalization would be more informative, since this would allow resolving potential differences between sgRNA and DBsgRNA delivery efficacies

We understand the point the reviewer is making but disagree with the idea to remove

normalization. There are positions in the backbone of the CRISPRoff sgRNA in which bases have been substituted with the photocleavable spacers. Because of this, reverse transcriptase has difficulty processing through the entire CRISPRoff sgRNA and is thus always “lower” than standard sgRNA. This is evident when running *in vitro* comparisons using known copy numbers of sgRNA and CRISPRoff. Unfortunately, this method does not give us a means to compare potential differences in deliveries. However, we believe that the similarities in editing percentage demonstrates that differences in delivery are negligible.

10) Fig. S2a: Why does editing in the sgRNA control samples decrease with prolonged UV exposure?

The reviewer has accurately identified a mistake in our data analysis and misattribution of samples. We have removed this figure and instead replaced with a figure showing the effect of length of exposure on DBsgRNAs. Equivalent editing results with or without light can be seen in figure 2 using standard sgRNAs.

11) What does “Mod” refer to in Fig. S1? Also, I do not understand the origin of the many bands in Fig. S1A. Should there not be only two bands visible in the illuminated samples? The authors should consider providing a scheme of the different sgRNA designs and corresponding cleavage fragments, which would highly simplify interpretation of the bands in Fig. S1A.

Mod in this figure referred to O-Me modified sgRNAs that have been synthetically made. We have removed this language for clarity.

Reviewer #3

In this paper, Stevermer et. al. present a method for controllable CRISPR using a guide RNA with a photocleavable linker. Overall, the work is well done and of general interest but with overstated results and lacking detail on experimental methods. I think if these things are fixed then it could be publishable in Nature Communications. My main comments (below) are related to adding experimental details, improving the figures, and not overstating results.

Major comments:

- Figure 2 a/b/c is VERY difficult to make sense of the way the left and right panels are plotted. I think you have to somehow merge left and right panels or think about a different way to plot your results that aren't so difficult to decipher. Also it would be useful if you can line up for the 3 cell lines the common targets (e.g. FANCF shifts right in 2c). Personal suggestion – I'd probably merge left and right into one with 4 bars at each target and remove the data points for clarity and include those in SI instead.

We thank the reviewer for the suggestion on rearranging the figure mentioned. We have attempted multiple different formats of presenting this data at a variety of conferences. From the feedback we have received, seeing all four bar graphs together muddles the message while separating the CRISPRoff and standard sgRNAs highlights the differences between the methodologies.

Overstatements that need repair:

- o “However, each of these strategies requires a new, non-native component of the CRISPR system.” First, this doesn't seem entirely true, and second how does yours also not fall into this category?

We intended for the characterization of non-native to be engineered protein complexes such as mentioned in the references to similar strategies. We have included language to make this distinction more apparent.

- o “We next demonstrated that DBsgRNAs are completely cleaved within cells upon illumination” – not “complete” RNA cleavage.

We have removed this absolute statement and replaced with softer language.

- o “intact DBsgRNAs formed DSBs at a similar frequency as standard sgRNAs”. This statement is neither quantitative nor easily verifiable from your data. It feels like you're trying to sweep under the rug the many cases where this is not true.

We have run a more rigorous statistical analysis across all targets and discovered that all but two sgRNAs do not behave significantly different than standard sgRNAs. We have added this to the discussion and detailed the statistical analysis that was performed.

- o “We determined that the decrease in efficiency is based on the individual editing kinetics at each site” – this appears to be an educated guess or a suspicion. You can either prove it or tone it down. Also as a counterexample EMX1 changes editing dramatically between cell lines when presumably the kinetics of editing probably doesn't.

The reviewer brings up an excellent point about the assertion of individual editing kinetics. To address this we have performed a high-resolution analysis on the FANCF site that appears to edit rapidly. We have demonstrated that by addition of light 15 minute post-transfection we are able to obtain control of the editing events. These data can be seen in Fig S2e.

We disagree that the editing kinetics are similar across cell lines as local chromatin context and expression patterns have been shown to influence editing outcomes (Verkuijl 2019). Further we

have seen that the indel profile between cell lines can be significantly different as seen in the data below.

In HEK (sample 103) +1 insertions are very common. However, in U2OS cells (102) the -6 deletion is frequently observed. This genotype that is not seen in HEK.

o “One of the major advantages of using optical opposed to chemical stimulus is the ability to obtain precise spatial control.” – This should have an explanation. I could see an advantage in tissues, but then you run into low UV transmission and potential tissue damage that will probably make this method not useful. You have a nice demo of spatial control, but you need to give examples of what it could be used for. I personally can’t think of one for cells, given that you could always physically split them instead.

We have added further discussion to address the advantages and benefits of tight spatial control in cell culture as well as potentially *in vivo* experiments.

o “We have successfully demonstrated this technology in multiple human cell lines across multiple genomic loci and expect this technology to be compatible with any CRISPR-based application. “ - you forgot the “m” on many.

While we appreciate the turn-of-phrase supplied by the reviewer, we do believe that this technology is compatible with any CRISPR technology that uses exogenously produced sgRNAs.

- Fragmentation

o Report estimated band lengths in FigureS1A.

We have included the ladder lengths on the figure and expanded on the associated figure legend to make clear that the two fragment sizes will add to the full-length product.

o The full sequence and length of your sgRNAs are not reported anywhere.

We have included a new figure S1c that has the entire sgRNA sequence as well as PC-linker locations .

Missing details on synthesis/purchase of photocleavable RNA

We have included further details on how CRISPRoff sgRNAs are synthesized.

o Why are there sometimes two and sometimes 3 fragments for a single cleavage location?
We assume that the reviewer is referring to Fig. S1e. Two cleavage products are a result of asymmetrical division within the sgRNA, giving rise to two distinct RNA fragments on the gel. The third band is the remaining full length product within these *in vitro* studies.

o FigureS1–Abbreviation are given without any explanation or context(“mod” “DNMT1” “FANCF” “VEGFA”)

We have removed all reference to ‘mod’ within the manuscript and stylized gene names in accordance with standard practice. Further language has been added to clarify these are genes.

o Experimental details of the fragmentation assay and analysis are not provided.
Additional details for the fragmentation assay have been added.

- Figure 1

o Panel b - “minor products match predicted weights based on cleavage locations” is not at all obvious, or verifiable based on the information we’re given. It appears a ~33 kDa product (with a ~17 kDa unidentified sub-product) degrades into fragments of ~24, 18, 14, 8. Based on my crude calculations I’d expect fragments of ~19, 9, and 6 kDa. Please resolve this. Also, visually the arrows are not descriptive on their own.

We have gone into further detail describing each minor product within the ESI traces. This includes the cases where only one linkage is broken, giving rise to a complex mixture of RNA. Each peak identified has been described along with approximate molecular weight. The 17kDa unidentified sub-product was an artifact of ESI deconvolution. We have rerun these traces to reduce any misinterpretation.

o Panel c – This experiment is not described in (any) sufficient detail. After sometime I get what is going on but it is missing 1) experimental protocol, 2) target dsDNA info, 3) sgRNA info (full length and fragment).

We have greatly expanded on the experimental detail in this assay. The intent is not to demonstrate cleavage of the sgRNA but rather that cleaved CRISPRoff sgRNAs are unable to create DSB breaks in dsDNA containing the protospacer+PAM sequence

o Paneld+e

§ Similar to c, target dsDNA and sgRNA sequences are not given.

Panel E refers to ddPCR showing abundance of sgRNA present within the cells. A cartoon in Fig. S1c has been added to show the relevant primers and their relation to the sgRNA backbone.

sgRNA target sequences can be found in Supp. Table 1. We have now included the exact target used in Fig. 1E

§ something should be added to visually distinguish these two. d shows RNA cleavage and e shows % editing in cells. A short title for each or a cartoon could help.
A small title has been added to these two subfigures.

o Panel f – I don’t see how this shows controlled gene editing without a proper control. The x-axis doesn’t make sense – CRISPR Activity with units of hours?

We have modified the X-axis to further illuminate the intent of the experiment and added a dashed line at the experimental endpoint to emphasize that all samples were harvested at the

same time, with the only difference between groups in the experiment being the time of light exposure post-transfection. We have added further details of the experiment within the manuscript.

- Illumination

o Is UV going to be a problem for passing through biological materials? Also UV is known to damage DNA, so maybe you can make some comments on those.

We have included a small discussion on the effect of UV on biological materials in the relevant sections. While we are keen to expand into longer wavelengths to avoid these effects, there is currently no chemistry to the best of our knowledge that provides these capabilities. We anticipate that as the chemistry progresses, we can incorporate these advances into future versions of CRISPRoff.

o Why broad spectrum light in Figure S2a? I think you mean broad spectrum UV, but it is not explained. The wavelength ranges should be stated. What is the source?

We have included a new figure (Fig. S1d) that shows the wavelengths present in the light source used. We have provided further detail in the methods regarding the exact light source.

o Since there is no difference between 345 and 355 filters in S1b, how is the 355 filter an optimization?

We agree with this reviewer that this was a negligible difference and have removed experiments pertaining to the usage of different filters.

o Why do you use 385 nm in Figure 2? Choice is not obvious and not explained.

385nm is the wavelength of a standard epifluorescent microscope using an LED light source. This was chosen because it is an instrument that many labs may have standard readily available for use. This has been expanded on.

Minor comments:

- ICE – unabbreviate at first instance and give brief description.

Full abbreviation has been written out and relevant paper cited.

- “indel” describe/unabbreviate at first use.

Full abbreviation has been written out.

- S2c would be clearer if you used 1 panel with two lines for DBsgRNA and sgRNA

We appreciate the thought the reviewer has given this panel. However, when we overlay the two traces, they are nearly identical and difficult to distinguish without significant effort.

- I would not use hn as a shorthand for light. First it only saves you 3 letters, second I don't think everyone will get it. Also, you write “v” incorrectly for the Greek nu in many instances, further confusing it.

We have replaced all instances with 'light'

- Figure 2e is distracting (and the result is obvious from 2f) – I would personally replace it with some version of S2f to show why the pattern emerges.

We disagree with the reviewer that this result is obvious. From the diagram of the fluorescent absorbance (Fig. S1b) it is not immediately apparent that a 385nm LED would cleave the CRISPRoff sgRNA. Further we believe that this is the most common use case for the CRISPRoff system as not every lab will have a custom built light source but rather most may prefer to use a microscope that they already have in-house.

- "OT1" and "OT2" abbreviations are not explained anywhere (Figure 2).

We have expanded the nomenclature in the legend to off-target site 1 and off-target site 2.

Reviewers' Comments:

Reviewer #1:

Remarks to the Author:

The revised manuscript is much improved, and essentially all of my most critical concerns have been addressed. The responses to the other reviewers also seem to be sufficient. This manuscript is suitable for publication, and the work remains as interesting and valuable as I stated in my initial review.

The inclusion of the additional information on the Cas9 protein being used is appreciated. I would strongly encourage the authors to add slightly more detail to their description of the Cas9 being used. My best attempts to identify the features of the protein sourced from Aldevron suggest that the *S. pyogenes* construct has an SV40 NLS sequence fused to each terminus of Cas9, e.g. [SV40 NLS][Spy Cas9][SV40 NLS]. It would be ideal to include this info in the materials & methods so that this important information is not lost in case, for example, Aldevron eventually starts selling SpyCas9 constructs with more/different NLS configurations.

<https://www.aldevron.com/hubfs/files/Resources%20files/ald-Cas9%20Product%20Guide-0419.pdf>

I am admittedly being a stickler, but the closing sentence "We anticipate that the CRISPRoff system will be a valuable tool for both in vitro and in vivo control of CRISPR." would benefit from the inclusion of any word after "CRISPR". CRISPR refers to a prokaryotic immune system, and here the authors are clearly referring to CRISPR-based tools, which calls for phrases such as CRISPR technologies, CRISPR tools, etc.

Reviewer #2:

Remarks to the Author:

The manuscript improved substantially during revision. The revised manuscript describes the DBsgRNA design and synthesis in the necessary detail and now contains the relevant information to both assess and independently reproduce the reported findings. I have no doubt that DBsgRNAs will be a very need tool to control genome editing and appreciated by the CRISPR community. Below, I made some additional comments and suggestions. Once these remaining issues - most of which are minor - are resolved, I gladly support publication of this interesting study in Nature Communications.

1. Figure 2: In the legend, it says that four experimental replicates were performed. However, only three, sometimes even just two datapoints (e.g. Fig. 2b, right, STK3_sg1 sample; left PRKAG3 sample) are shown for each condition. Please add missing datapoints/correct.
2. Similar in Figure S3b: For FANCF, MIP, several curves are interrupted/datapoints seem to be missing.
3. Figure S1g: For me, it was not intuitive, that the numbers indicate size in Da. I suggest rounding numbers and stating that these indicate kDa, e.g. "32 kDa" instead of "32300"
4. Hoechst is misspelled ("Hoescht") in Fig. S2d and Fig. 3d
5. Wording in Line 54: "Activity of the sgRNA was qualified by detection of PCR fragments comprising of the full-length PCR product through a fragment analyzer." It took me a while to get what is meant here. Consider simplifying, e.g. "We investigated sgRNA activity by assessing target DNA cleavage with a fragment analyzer".

6. Line 56: "Importantly, when DBsgRNAs were synthesized as fragments and mixed with SpCas9, they did not exhibit any editing activity when assessed by the same assay". I would not necessarily talk about "editing" in context of in vitro cleavage, but simply "cleavage".

7. Line 76: better write "samples were illuminated for up to 60 seconds", since in the referenced figure (Fig. S2b), also shorter illumination times were used.

8. Line 81-85: "Editing in illuminated sgRNA RNP populations was not significantly different than paired populations left in the dark. The lack of difference in overall editing efficiency following illumination compared to dark with sgRNA RNPs suggests..." Sounds unnecessarily complicated. Consider revising

9. Line 135: replace "off-target effects" by "off-target sites"

10. Line 145: consider writing just "laser" instead of "laser line"

11. Line 149: "... microscope setup contained a 385 nm laser line" Did you really use a laser here? In the methods it says that the microscope was equipped with a 385 nm LED, not a laser.

12. Line 164: "Further, because CRISPRoff makes modifications to the backbone of the sgRNA it can be compatible with other technologies, such as sgRNA modifications to activate gene editing²⁵...". I acknowledge the point made here by the authors. However the example cited here, i.e. reference 25, seems rather suboptimal. Ref. 25 reports caged sgRNAs that are activated by the identical wavelength of light, i.e. UV light. How should this concept be compatible with the author's CRISPRoff technology? Incorporating o-nitrobenzyl photocleavable linkers into the photocaged guides developed by Zhou et al. would simply result in sgRNAs that are impaired in the absence of light (due to photocaging) and fragmented in the presence of UV (due to CRISPRoff), i.e. they would be inactive irrespective of irradiation.

Reviewer #3:

Remarks to the Author:

The authors have done a good job in addressing my concerns, and I think the paper is suitable for publication.

One caveat is that I still strongly dislike the presentation in figure 2. I appreciate there is a lot of data there and there are tradeoffs with presenting in different ways but I don't think Figure 2 is easy to digest and get the information that you intend the reader to get. I still strongly suggest alternate presentation but I acknowledge this is just my opinion and the manuscript is solid with or without such change.

Reviewer #1 (Remarks to the Author):

The revised manuscript is much improved, and essentially all of my most critical concerns have been addressed. The responses to the other reviewers also seem to be sufficient. This manuscript is suitable for publication, and the work remains as interesting and valuable as I stated in my initial review.

The inclusion of the additional information on the Cas9 protein being used is appreciated. I would strongly encourage the authors to add slightly more detail to their description of the Cas9 being used. My best attempts to identify the features of the protein sourced from Aldevron suggest that the *S. pyogenes* construct has an SV40 NLS sequence fused to each terminus of Cas9, e.g. [SV40 NLS][Spy Cas9][SV40 NLS]. It would be ideal to include this info in the materials & methods so that this important information is not lost in case, for example, Aldevron eventually starts selling SpyCas9 constructs with more/different NLS configurations.

<https://www.aldevron.com/hubfs/files/Resources%20files/ald-Cas9%20Product%20Guide-0419.pdf>

The materials and methods have been updated to state SV40 NLS as suggested.

I am admittedly being a stickler, but the closing sentence "We anticipate that the CRISPRoff system will be a valuable tool for both in vitro and in vivo control of CRISPR." would benefit from the inclusion of any word after "CRISPR". CRISPR refers to a prokaryotic immune system, and here the authors are clearly referring to CRISPR-based tools, which calls for phrases such as CRISPR technologies, CRISPR tools, etc.

We do not think the reviewer is a stickler but rather correctly identifies an overly used shorthand. We have updated "CRISPR" to "CRISPR technologies".

Reviewer #2 (Remarks to the Author):

The manuscript improved substantially during revision. The revised manuscript describes the DBsgRNA design and synthesis in the necessary detail and now contains the relevant information to both assess and independently reproduce the reported findings. I have no doubt that DBsgRNAs will be a very need tool to control genome editing and appreciated by the CRISPR community.

Below, I made some additional comments and suggestions. Once these remaining issues - most of which are minor - are resolved, I gladly support publication of this interesting study in Nature Communications.

1. Figure 2: In the legend, it says that four experimental replicates were performed. However, only three, sometimes even just two datapoints (e.g. Fig. 2b, right, STK3_sg1 sample; left PRKAG3 sample) are shown for each condition. Please add missing datapoints/correct.

We thank the reviewer for noticing this oversight. They are correct that three experimental replicates were performed. This has been updated.

2. Similar in Figure S3b: For FANCF, MIP, several curves are interrupted/datapoints seem to be missing.

The previous version of this figure did not display the difference between a 'dark' sample and a sample that was illuminated at the same time point as these are functionally the same. This figure has been updated to include this point to make curves complete.

3. Figure S1g: For me, it was not intuitive, that the numbers indicate size in Da. I suggest rounding numbers and stating that these indicate kDa, e.g. "32 kDa" instead of "32300"

We agree that this would greatly improve readability. This change has been made.

4. Hoechst is misspelled ("Hoescht") in Fig. S2d and Fig. 3d

This has been updated in both relevant figures.

5. Wording in Line 54: "Activity of the sgRNA was qualified by detection of PCR fragments comprising of the full-length PCR product through a fragment analyzer." It took me a while to get what is meant here. Consider simplifying, e.g. "We investigated sgRNA activity by assessing target DNA cleavage with a fragment analyzer".

We agree that this is language is much more clear and have updated the manuscript accordingly.

6. Line 56: "Importantly, when DBsgRNAs were synthesized as fragments and mixed with SpCas9, they did not exhibit any editing activity when assessed by the same assay". I would not necessarily talk about "editing" in context of in vitro cleavage, but simply "cleavage".

The reviewer is correct that this is more accurate nomenclature. This has been updated.

7. Line 76: better write "samples were illuminated for up to 60 seconds", since in the referenced figure (Fig. S2b), also shorter illumination times were used.

This point is well received. 60 seconds was chosen as it robustly removed sgRNA activity as demonstrated by the referenced figure.

8. Line 81-85: "Editing in illuminated sgRNA RNP populations was not significantly different than paired populations left in the dark. The lack of difference in overall editing efficiency following illumination compared to dark with sgRNA RNPs suggests..." Sounds unnecessarily complicated. Consider revising

To increase readability, we have updated these lines as follow:

"Editing in illuminated sgRNA RNP populations was not significantly different than paired populations left in the dark. The similarity in overall editing efficiency following illumination when using standard sgRNA RNPs suggests that DBsgRNAs were effectively cleaved within cells and no longer functional."

9. Line 135: replace "off-target effects" by "off-target sites"

This has been replaced

10. Line 145: consider writing just "laser" instead of "laser line"

The word line has been removed.

11. Line 149: "... microscope setup contained a 385 nm laser line" Did you really use a laser here? In the methods it says that the microscope was equipped with a 385 nm LED, not a laser.

We thank the reviewer for noticing this important point. We did use a 385nm LED. This setup provides illumination to a much greater area. This has been corrected.

12. Line 164: "Further, because CRISPRoff makes modifications to the backbone of the sgRNA it can be compatible

with other technologies, such as sgRNA modifications to activate gene editing²⁵...". I acknowledge the point made here by the authors. However the example cited here, i.e. reference 25, seems rather suboptimal. Ref. 25 reports caged sgRNAs that are activated by the identical wavelength of light, i.e. UV light. How should this concept be compatible with the author's CRISPRoff technology? Incorporating o-nitrobenzyl photocleavable linkers into the photocaged guides developed by Zhou et al. would simply result in sgRNAs that are impaired in the absence of light (due to photocaging) and fragmented in the presence of UV (due to CRISPRoff), i.e. they would be inactive irrespective of irradiation.

At the moment the majority of photocleavable chemistry is constrained to the UV end of the light spectrum. However, we do acknowledge that there is work ongoing to shift the reactive window of these molecules so that the spectrum overlaps. We cite this reference to give the reader an idea of what might be possible in the future with further research and development.

Reviewer #3 (Remarks to the Author):

The authors have done a good job in addressing my concerns, and I think the paper is suitable for publication.

One caveat is that I still strongly dislike the presentation in figure 2. I appreciate there is a lot of data there and there are tradeoffs with presenting in different ways but I don't think Figure 2 is easy to digest and get the information that you intend the reader to get. I still strongly suggest alternate presentation but I acknowledge this is just my opinion and the manuscript is solid with or without such change.

We thank the reviewer for their acknowledgement of the work. To their suggestion, since submitting for review we have tried to reformat Figure 2 in a multitude of ways from only box-and-whiskers to breaking apart on the basis of sgRNA or light exposure to moving to separate graphs for each condition. Each of these has brought confusion to the audience we present to.